# A NanoBiT assay to monitor membrane proteins trafficking for drug discovery and drug development

Arfaxad Reyes-Alcaraz [1✉], Emilio Y. Lucero Garcia-Rojas[1], Elizabeth A. Merlinsky[1], Jae Young Seong[2], Richard A. Bond[1] & Bradley K. McConnell [1✉]

Internalization of membrane proteins plays a key role in many physiological functions; however, highly sensitive and versatile technologies are lacking to study such processes in real-time living systems. Here we describe an assay based on bioluminescence able to quantify membrane receptor trafficking for a wide variety of internalization mechanisms such as GPCR internalization/recycling, antibody-mediated internalization, and SARS-CoV2 viral infection. This study represents an alternative drug discovery tool to accelerate the drug development for a wide range of physiological processes, such as cancer, neurological, cardiopulmonary, metabolic, and infectious diseases including COVID-19.

[1] Department of Pharmacological and Pharmaceutical Sciences, College of Pharmacy, University of Houston, Houston, TX 77204-5037, USA. [2] Korea University, College of Medicine, Anam-dong, Seongbuk-gu, Seol 136-701, Republic of Korea. ✉email: areyesa2@central.uh.edu; bkmcconn@central.uh.edu

Membrane receptors participate in the detection of various environmental stimuli. Their internalization follows the functional process of receptor binding to agonists[1]. Receptor trafficking is also part of cell signaling[2]. Alterations in receptor trafficking have been reported in neurological diseases such as human startle disease hyperekplexia[3–6]. Also, alterations in the rate of constitutive degradation and on the trafficking of GABA receptors in epilepsy have been reported[7]. Membrane receptor trafficking is also seen during viral infection[8], one good example is the molecular mechanism of how the Human Inmunodeficiency Virus (HIV) enters the white blood cells through chemokine receptor 4 (CXCR4) and 5 (CCR5)[9]. In cardiovascular diseases[10] we have the case of the internalization of angiotensin II type 1 receptor ($AT_1R$) which plays an important role in maintaining cardiovascular homeostasis. Decreased internalization of $AT_1R$ is closely related to hypertension, induced by abnormal activation of this receptor[11].

Antibody-drug conjugates (ADCs)[12] are an excellent therapeutic approach in cancer therapy because they can directly bind to cancer cells and deliver the drug to kill them. In this case, the target membrane receptors are required to be highly and selectively expressed on cancer cells but also susceptible to be internalized by antibody binding[13]. To date, some studies of membrane proteins internalization have been reported[14–19]. However, some of the methodologies that monitor internalization have been costly and limited to a specific internalization mechanism[20–23]. In some cases, to monitor physiological processes such as receptor internalization, it is necessary to label the components that participate in such a process. The first labeling agents used to monitor receptor internalization were radioactive tags[24–26]. Later on, cell-based assays employing fluorescence were developed. However, such technologies require the use of expensive equipment; such as gamma radiation counters and confocal microscopes, and costly conjugated antibodies with a fluorophore[27,28]. Recent studies reported in the literature have started to use bioluminescence to study receptor internalization[29].

In this work, we hypothesized that, due to the high sensitivity of bioluminescence[30,31], we could devise a new assay using blue light emission to monitor a wide variety of internalization processes by targeting the early endosomes and using NanoLuc Binary Technology (NanoBiT)[32]. Bioluminescence resonance energy transfer (BRET) between a Renilla luciferase-inserted GPCR and a GFP10-fused FYVE domain was previously developed to measure GPCR internalization[33]. Therefore, we replaced the BRET pair with a split luciferase (NanoBiT) and we added flexible amino acid linkers in between (Supplementary Figs. 1 and 2). Here we report a methodology based on bioluminescence, produced by the fragment complementation of Nano Luciferase (NLuc). This assay allows quantifying in real-time the membrane protein internalization and recycling in living systems by using bioluminescence. This method can be applied to monitor the internalization of several types membrane proteins and contribute to elucidate receptor trafficking mechanisms as well as in vitro drug screening.

## Results

**GPCR trafficking.** We initially focused on studying the different internalization kinetics of class A G-protein coupled receptors (GPCRs). For this, we first established a set of GPCRs involved in a wide variety of physiological functions and diseases such as Galanin (GAL receptors), Chemokine receptors (CCR), and β-adrenergic receptors (βAR). We devised a strategy to monitor how the GPCR bound to its corresponding ligand is removed from the cell surface, subsequently localized in early endosomes, and then finally recycled into the cell membrane. To accomplish this, we covalently linked a small fragment of NLuc (SmBiT) to the C-terminal of the receptor, where a flexible linker is located between the two proteins (Supplementary Figs. 1, 2). This linker was designed to be composed mainly of glycine and alanine amino acids to provide flexibility and not to alter the pharmacological properties of the native receptor (Fig. 1a, b).

To detect light emission when the receptor is being localized at the early endosomes, we focused on the interaction between the early endosome and the FYVE zinc finger domain during the receptor trafficking; where the FYVE zinc finger domain is named after the four cysteine-rich proteins: **F**ab 1 (yeast orthologue of PIKfyve), **Y**OTB, **V**ac 1 (vesicle transport protein), and **E**EA1(Early Endosome Antigen 1). FYVE domains bind phosphatidylinositol 3-phosphate from early endosomes, in a manner that is dependent on its metal ion coordination and basic amino acids. This FYVE domain inserts into cell membranes in a pH-dependent manner[34], where it is composed of two small beta hairpins (or zinc knuckles) and followed by an alpha helix. Additionally, the FYVE finger binds two zinc ions where this FYVE finger has eight potential zinc coordinating cysteine positions and is characterized by having basic amino acids around the cysteines[35]. To achieve this, the gene of the FYVE domain of the human Endofin (residues from Q739 to K806) was synthesized and covalently attached by molecular cloning into a large fragment of NLuc (LgBiT) at the N-termini (Fig. 1a, b).

We observed an increase in luminescence in the β-adrenergic receptor 2 (β2AR) and two additional GPCRs upon agonist-mediated stimulation (Fig. 1c), reaching a maximum in luminescence intensity at 30 min after agonist treatment. In the case of β2AR, we used epinephrine to cause epinephrine-mediated sequestration of receptors from the plasma membrane and translocate them into early endosomes. This increased the blue luminescent signal in a concentration-dependent manner in response to the agonist (Fig. 1d). Regarding the quantitative pharmacological analysis of GPCRs, HEK293 cells were treated with increasing concentrations of agonists. The time course graph displayed an increase in normalized luminescence over time with increasing agonist concentrations. The curve analysis of this response demonstrates a clear concentration-dependent response of human β2AR internalization (Fig. 1d, f).

To show the specificity of the internalization processes, we then validated our approach by using inhibitors targeting the vital elements involved in receptor endocytosis (Fig. 1e). Internalization of a prototypical GPCR was inhibited by using the GPCR kinase 2/3 inhibitor (Cmpd101), a selective inhibitor of clathrin-mediated endocytosis (PiTStop) and dynamin inhibitor (Dynasore), consistent with the role that β-arrestins have in agonist-dependent and mediated internalization of GPCRs. Finally, a dose-response stimulation curve was obtained from (Fig. 1d) to quantify the internalization potency for the epinephrine at the β2AR (Fig. 1f).

**Assessing recycling and forward trafficking of a prototypical GPCR.** By using the same approach, we were able to study not only GPCR internalization but also receptor recycling (Fig. 2a). After treating the cells expressing β2AR-SmBiT with the final concentration of 10 μM epinephrine, the internalization process occurred. This internalization process was observed by a rapid increase in luminescence, indicating the receptor was being removed from the cell surface (Fig. 2, number-1). This was then followed by a maximum in the luminescent signal being reached, suggesting the receptor was localized at the endosome (Fig. 2, number-2). After a few minutes of signal stability, we took the plate out of the luminometer, removed the medium containing the ligand and replaced it with fresh medium in the absence of

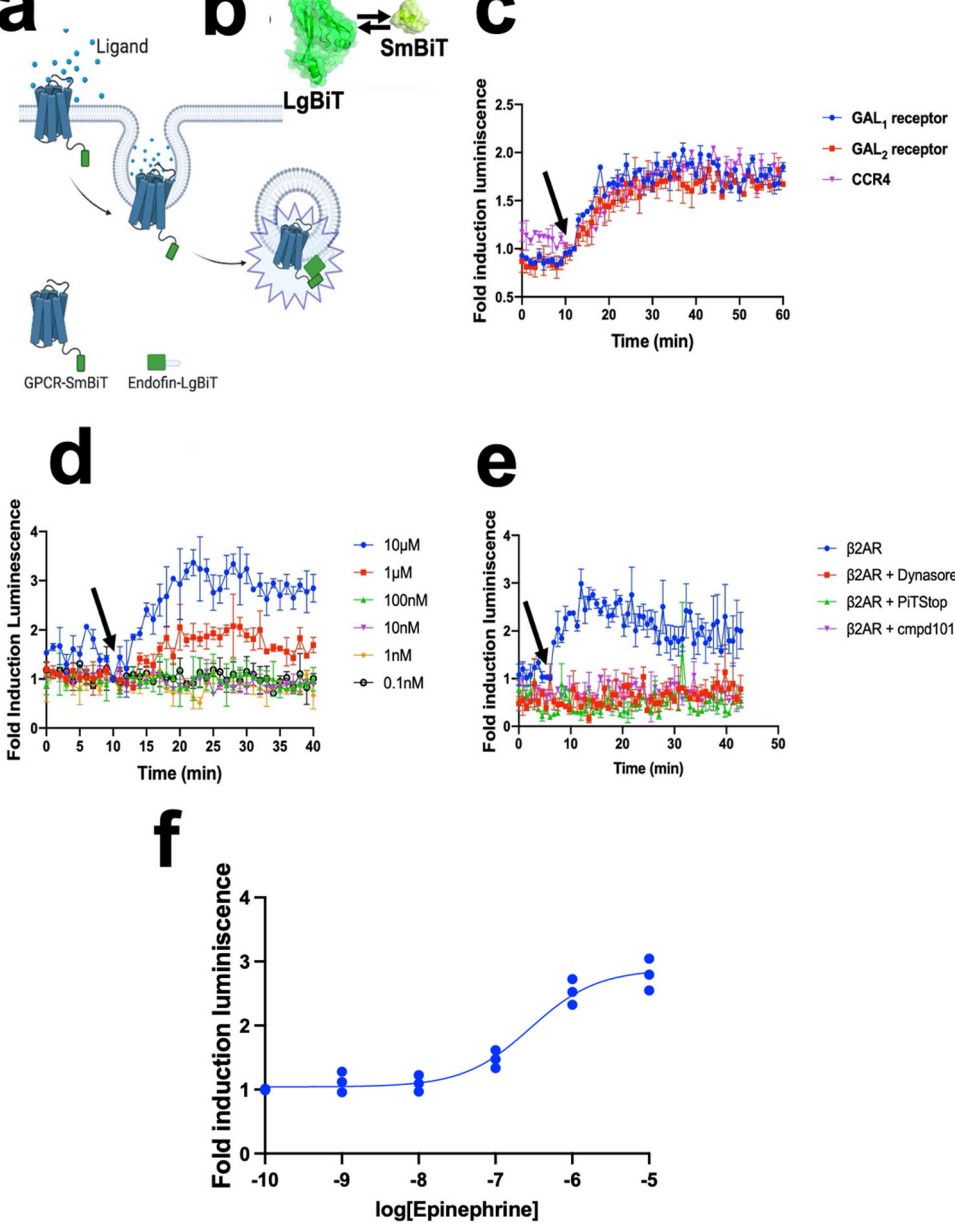

**Fig. 1 GPCR internalization via early endosomes. a** Schematic of the overall concept of our structural complementation assay to monitor internalization in real-time living cells. **b** Schematics of our structural complementation system based on Nano Luciferase showing the LgBiT (Large fragment of Nano Luciferase) and SmBiT (Small fragment of Nano Luciferase). **c** Different internalization rates across different GPCRs. Galanin receptors were stimulated using 10 µM Galanin (GAL$_1$ and GAL$_2$ receptors) whereas, in the case of chemokine receptor 4 (CCR4), 100 ng/ml of SDF-1α was used. **d** Dose-dependent internalization of human β2AR in early endosomes in cardiomyocytes using different concentrations of epinephrine. **e** Validation of the GPCR internalization assay using 10 µM of endocytosis inhibitors, cmpd101, PiTStop, and Dynasore. **f** Dose-response curve stimulation for internalization of β2AR. The results are expressed as mean ± s.e.m. of three experiments performed in triplicate; each triplicate was averaged before calculating the s.e.m. The corresponding Z' factor was 0.52.

epinephrine, and continued measuring the luminescence signal. We observed a gradual decay of signal in the absence of ligand, indicating the ligand–receptor complex was being dissociated and the receptor was localized back to the plasma membrane (Fig. 2, *number-3*). It is interesting to note that the β2AR recycling back to the cell surface was slightly slower as compared to its internalization kinetics.

**Luminescent signal is originated from early endosomes.** To demonstrate that the luminescent signal is generated from early endosomes, we performed a proximity ligation assay (PLA). The

PLA assay is a powerful tool to detect close proximity (about 30 nm) between two entities with high specificity and sensitivity[36–38]. In this case, the protein targets we used were (1) NLuc, attached to the FYVE domain, (2) EEA1, at the early endosome, and (3) the β2AR (Fig. 3). We then used two primary antibodies, raised in different species (rabbit and mouse), to detect two unique protein targets (β2AR and EEA1 or NLuc and EEA1). A pair of oligonucleotide-labeled secondary antibodies (PLA probes) were bound to the primary antibodies (Fig. 3a, d). Next, hybridizing connector oligos joined the PLA probes. If the PLA probes were in close proximity to each other and, the ligation process formed a closed circle, serving as

the DNA template required for the rolling-circle amplification (RCA). This allowed up to a 1000-fold amplified signal that was still tethered to the PLA probe, allowing localization of the signal. Lastly, labeled oligos hybridized to the complementary sequences within the amplicon which were then visualized and quantified as discrete red spots (PLA signals) by microscopy image analysis (Fig. 3c, f).

**Live cell imaging: GPCR internalization**. In order to visualize the GPCR internalization in real time living cells, we used a bioluminescent microscope to characterize the receptor localization within the cells. This trafficking visualization is unique in that being able to measure receptor internalization in living cells and in real-time by using luminescence. Currently, most other

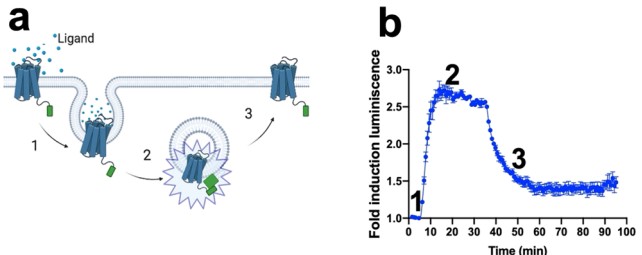

**Fig. 2 Monitoring in real-time the internalization and recycling of a prototypical GPCR. a, b** Recycling of β2AR in HEK293 cells using 10 μM epinephrine; Number "1" indicates the receptor being removed from the plasma membrane, Number "2" corresponds to the receptor being localized in early endosomes, and Number "3" indicates the washout of the ligand and return of the receptor to the cell surface. The arrows indicate the time at which the cells were treated with the corresponding ligand. The results are expressed as mean ± s.e.m. of three experiments performed in triplicate; each triplicate was averaged before calculating the s.e.m.

technologies that quantify receptor internalization do not offer information on spatiotemporal live-cell imaging since they are not performed in real-time and living systems. Here, we were able to visualize the trafficking of a prototypical GPCR (β2AR) in real-time and in living cells (Fig. 4). This trafficking was observed as small luminescent spots moving through the cytosol (highlighted in arrows). After ligand addition, we recorded the β2AR internalization by capturing total luminescence every 2 min. We observed ~25% decrease in bioluminescent signal intensity along the experiment (Supplementary Fig. 6), presumably as a consequence of furimazine (NLuc substrate) depletion.

**Non-GPCR membrane receptor internalization**. To continue exploring potential applications of our technology, we next focused on a prototypical non-GPCR receptor, the human epidermal growth factor receptor 2 (EGFR2), also referred to as the HER2 receptor (Fig. 5a). In this study, we were able to observe that the internalization rate was slightly slower compared to some GPCRs, suggesting that agonists that activate membrane receptors follow similar internalization kinetics (Fig. 5b).

**Antibody-mediated internalization**. Being able to measure antibody-mediated internalization is currently one of the most exciting receptor trafficking mechanisms to study in medicine. To test the versatility of our methodology, we set up a strategy to monitor antibody-mediated internalization. We used a membrane protein recently discovered termed FAM19A5 Isoform II (also called TAFA5). It has been described that FAM19A5 plays a key role in neurological disorders[39]. We decided to use FAM19A5 Isoform II as a prototypical system to study antibody-mediated internalization using two antibodies currently under development. We tagged FAM19A5 Isoform II with SmBiT at the N-termini. SmBiT-FAM19A5 isoform II was co-expressed with the FYVE domain

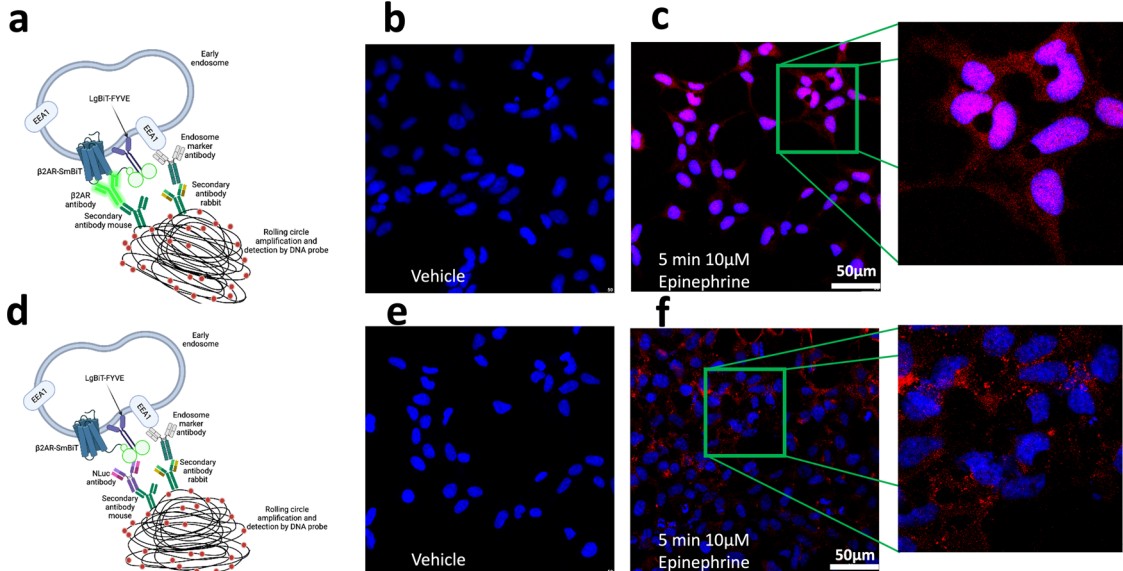

**Fig. 3 Schematic of the Proximity Ligation Assay (PLA).** Two primary antibodies recognize specific proteins in the cell: **a** β2AR-SmBiT and Early Endosome Antigen 1 (EEA1). **d** LgBiT-FYVE and Early Endosome Antigen 1. In both cases, secondary antibodies coupled with oligonucleotides (PLA probes) bind to the primary antibodies. When the PLA probes are in close proximity, connector oligos join the PLA probes and become ligated. The resulting closed, circular DNA template becomes amplified by DNA polymerase. Then, complementary detection oligos coupled to fluorochromes hybridize to repeating sequences in the amplicons. PLA signals are detected by fluorescent microscopy as discrete spots and provide the intracellular localization of the protein or protein interaction. Confocal images were obtained from the PLA of the early endosomes (**b**, **c**, **e**, **f**). Image **b** shows the PLA assay using antibodies targeting β2AR and EEA1 in the absence of ligand (vehicle control). Image **c** shows the PLA assay using the same antibodies as in image B but where the sample was incubated for 5 min in the presence of 10 μM epinephrine. Image **e** shows the PLA assay using antibodies targeting the LgBiT-FYVE and EEA1 in the absence of ligand (vehicle control). Image **f** shows the PLA assay using the same antibodies as in image e but where the sample was incubated for 5 min in the presence of 10 μM epinephrine. High magnification (scale bar = 50 μm) or image **c** and image **f** are shown in the inserts to the far right, respectively.

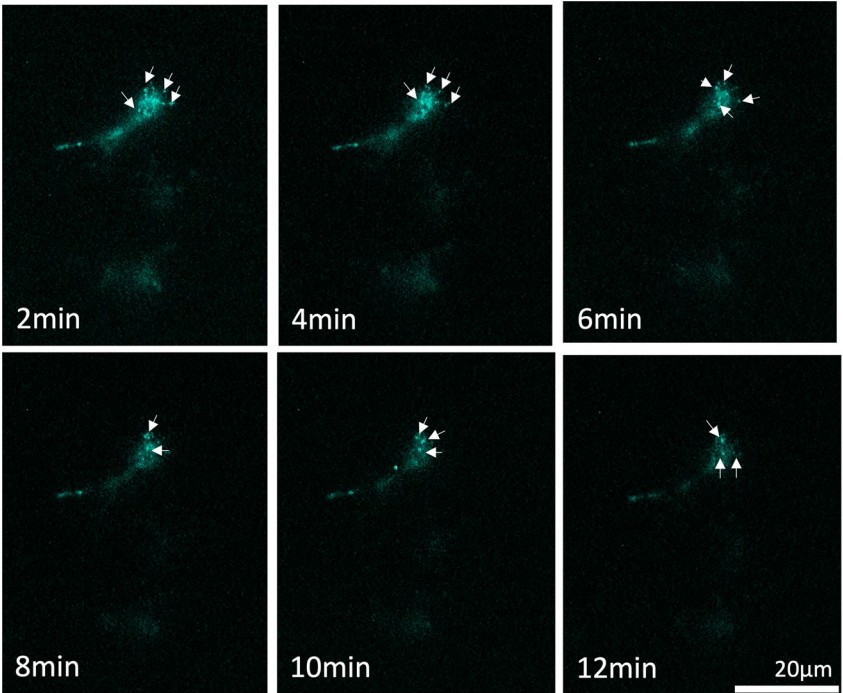

**Fig. 4 Visualization of receptor localization in HEK293 cells using a bioluminescence LV200 Olympus microscope.** HEK293 cells were transfected with β2AR-SmBiT and LgBiT-FYVE. The luminescence images were acquired after the addition of the luciferase substrate, furimazine, and 10 µM epinephrine (final concentration) by capturing total luminescence for 2 min at each of the indicated times. Scale bar represents 20 µm.

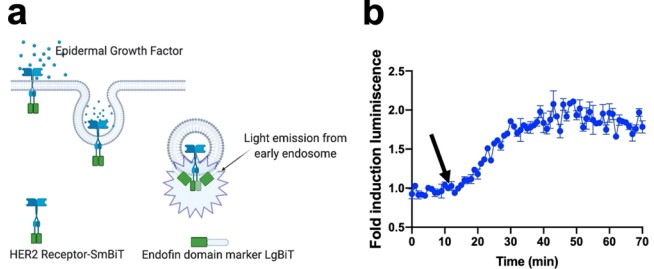

**Fig. 5 Real time monitoring of internalization of a prototypical non-GPCR. a** Schematic representation for monitoring internalization of a non-GPCR, the HER2 receptor. **b** HER2 receptor time course internalization of cells expressing the receptor treated with 10 µM (final concentration) human epidermal growth factor (hEGF). The arrow indicates the time when the cells were treated with the virus. The results are expressed as mean ± s.e.m. of three experiments performed in triplicate; each triplicate was averaged before calculating the s.e.m. The corresponding Z' factor was 0.76.

tagged with the LgBiT at the N-termini (Fig. 6a). In this experiment, we observed a slower internalization kinetics than observed for the GPCRs (Fig. 1). This slower internalization kinetics suggested a slow conformational change in the receptor induced by the binding with the antibody (Fig. 7b). We achieved to quantify and compare the internalization potencies for the two antibodies, highlighting that there is no statistical difference between them (Fig. 6c).

**SARS-CoV2 viral entry**. Finally, since internalization is also present in the process when a virus enters into the cell, we devised a simple system to monitor that molecular mechanism by adapting the SARS-CoV2 Spike protein to a lentiviral system as a model of viral infection (Fig. 7a). In this experimental approach, we observed viral entry into the cells is mediated by early endosomes and that its kinetics is also much slower as compared to ligand-activated receptors (Fig. 1). The viral entry trafficking,

using the SARS-CoV2 Spike protein (Fig. 7b), was similar to the antibody-mediated receptor trafficking (Fig. 6b, c). In ongoing experiments, we are exploring whether new variants (delta and Omicron) of SARS-CoV2, increase the internalization rate and kinetics of the Angiotensin-Converting Enzyme 2 (ACE2) internalization.

## Discussion

In this study, we described several strategies to accurately quantify membrane receptor internalization across different systems by setting up a structural complementation assay based on NLuc. Other studies have been reported in the literature that also used the structural complementation of NLuc but adapted to different physiological contexts such as GPCR dimerization and oligomerization[40,41]. The approach of structural complementation has great potential in drug discovery and structural biology. It has been successfully applied for several applications in biological research. One of the most interesting applications has been reported by Duan and colleges[42]. In this report, they used HiBiT technology (another structural complementation approach of NLuc). They used the strong affinity that exists between the two fragments of NLuc to stabilize a GPCR protein complex; the vasoactive intestinal polypeptide receptor and the heterotrimeric G-protein that were used for posterior structural studies involving Cryo-EM. Finally, another interesting work was made by Inoue and co-workers[43] where they designed a dissociation assay, to follow in real-time, the activation of Gα protein by monitoring its dissociation from Gβγ subunits and observed by a decrease of bioluminescence.

At the beginning of our study, we aimed to develop a drug discovery tool that can be applied to several types of receptors susceptible to undergo internalization. We hypothesized that it would be possible to monitor the activity of a particular receptor by observing in real-time, its trafficking in living systems. In this regard, we continued exploring different membrane receptor's

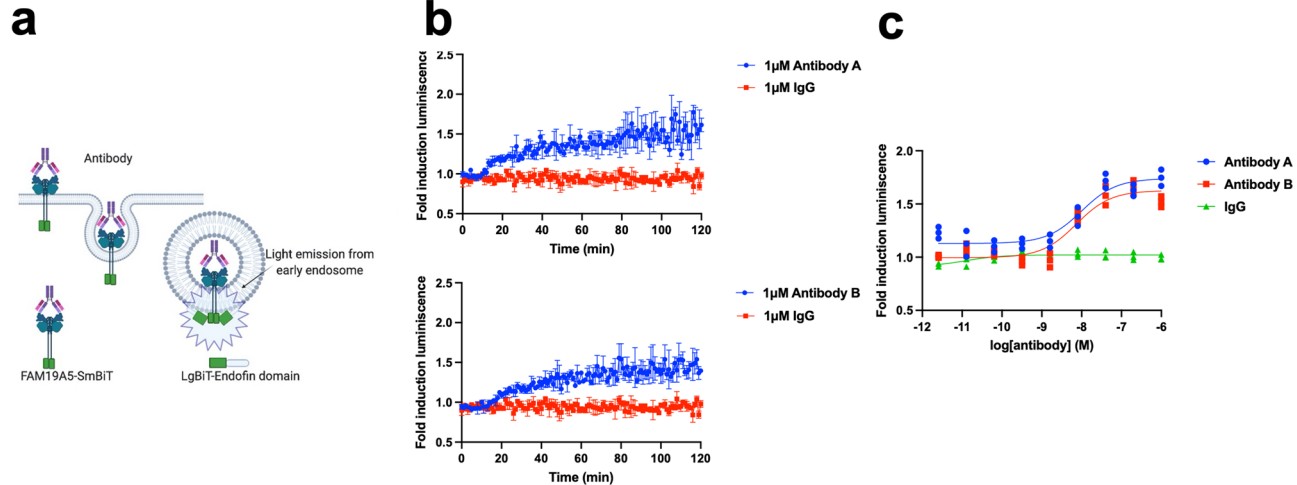

**Fig. 6 Membrane protein internalization by binding with two monoclonal antibodies. a** Schematic of a membrane protein being internalized by the binding with an antibody via early endosomes. **b** The membrane protein (FAM19A5 Isoform II) being internalized when cells expressing FAM19A5 Isoform II were treated with 10 and 100 nM of two antibodies (these antibodies are currently under development). 1 μM IgG antibody was used as a control. The *top panel* shows antibody A and the *bottom panel* shows antibody B. **c** Dose–response curves for both antibodies. The corresponding EC50 value for antibody A was 7.69 ± 0.30 nM (Z' factor of 0.63) and for antibody B (Z' factor of 0.64) was 8.15 ± 0.43 nM. The arrow indicates the time when the cells were treated with the virus. The results are expressed as mean ± s.e.m. of three experiments performed in triplicate; each triplicate was averaged before calculating the s.e.m.

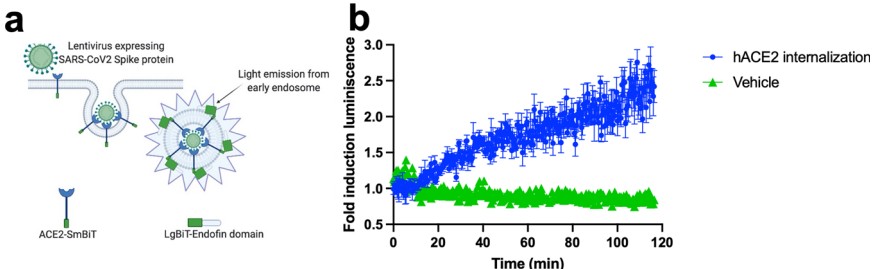

**Fig. 7 Monitoring SARS-CoV2 infection in real-time in living HEK293 cells. a** Schematic representation of how the Lentivirus expressing the SARS-CoV2 Spike protein is internalized by binding with Angiotensin-Converting Enzyme 2 (ACE2) via early endosomes. **b** Real-time monitoring of viral entry into HEK293 cells expressing human ACE2 when treated with the Lentivirus. The arrow indicates the time when the cells were treated with the virus. The results are expressed as mean ± s.e.m. of three experiments performed in triplicate; each triplicate was averaged before calculating the s.e.m. The corresponding Z' score was 0.55.

systems to extend the potential of our assay in studying membrane protein internalization induced by external stimuli.

For this purpose, we used the FYVE domain of endofin, since this domain selectively binds to early endosomes. We covalently linked the FYVE domain to the large fragment of NLuc (Supplementary Fig. 1). As the first application of our methodology, we decided to explore how internalization occurs across the class A GPCRs; the largest class of GPCRs, accounting for nearly 85% of the GPCR genes that encode for rhodopsin-like receptors (i.e., β2AR), olfactory and orphan receptors[44]. We were able to observe that the internalization kinetics reached a maximum within about 10 min after ligand stimulation and the fold induction (previously normalized against vehicle) was around 2, depending on the GPCR (Fig. 1c).

To validate our methodology, we pretreated the cells expressing our system with internalization inhibitors (Fig. 1e). We were able to verify that the luminescent signal was abolished in the presence of those inhibitors. Such observation suggests that the increase in the luminescent signal after ligand stimulation is originated from the GPCR internalization process.

One of the advantages of real-time assays is that we can study different conditions in the same well of the assay plate. To illustrate this, we monitored the GPCR internalization and

recycling in the same sample (Fig. 2a, b). On the other hand, we consider this technique useful in the de-orphanization of GPCRs, especially in cases where the receptor signaling is unknown, as well as in the development and characterization of agonists and antagonists for GPCRs. However, since the current development of our assay relies on transient transfections, the expression levels of the receptors are not at endogenous levels. Therefore, future directions will focus on developing assays that will express receptors at endogenous levels by using CRISPR gene editing techniques; thus, being able to better mimic the physiological conditions.

The potential application of this internalization assay is that it can be applied to other classes of membrane receptors beyond the study and characterization of the GPCRs, where GPCRs account for approximately 35% of the Federal Drug Administration (FDA) approved drugs[45]. In addition to GPCRs, we set out to apply our assay to receptor tyrosine kinases (RTKs), another class of membrane receptors. Specifically, we applied our methodology to the human epidermal growth factor receptor 2 (HER2), a member of the receptor tyrosine-protein kinase ErbB family of receptors that promote cell proliferation and opposes apoptosis[46]. The application of our technology to the HER2 receptor is highly significant since HER2 is known to have therapeutic importance

in breast and ovarian cancers[47]. Thus, we studied HER2 receptor internalization, where this membrane receptor trafficking pattern represents pivotal internalization steps towards the development of treatments of HER2 positive cancer patients[19]. Our results demonstrate that we can monitor membrane protein trafficking of RTKs, specifically HER2, using an approach similar to the methodology described for GPCRs.

As further validation regarding to the versatility of our membrane protein internalization assay, we applied this technology to monitor antibody-mediated internalization. As such, we studied the Family with sequence similarity 19 (chemokine (C-C motif)-like) member A5 (FAM19A5) receptor protein, a member of the TAFA family a chemokine-like protein that regulates cell proliferation and migration. Specifically, FAM19A5 is a novel gene with multiple physiological functions (i.e., neurokine, adipokine) recently being discovered[39,48]. For this membrane protein, we were able to study antibody-mediated internalization of FAM19A5 by two newly develop monoclonal antibodies (Fig. 6a, b). In contrast to GPCRs or RTK receptors, antibody-mediated internalization of FAM19A5 displayed much slower kinetics, reaching a maximum of internalization at two hours after antibody treatment. The fold induction was slightly lower than previously seen in GPCRs or RTK receptors, presumably because the binding between the antibody and FAM19A5 induced minor conformational changes on the receptor as compared to the ligand–receptor interactions observed for $\beta 2AR$ and HER2.

Finally, we wanted to extend our membrane protein internalization assay to monitor viral infections. Specifically, we set out to monitor SARS-CoV2, the seventh coronavirus known to infect humans and able to cause severe respiratory disease, can cause multiorgan infection and cell tropism in the human body[49–51]. Moreover, the SARS-CoV2-mediated receptor trafficking can be studied and characterized for drug discovery. Thus, when SARS-CoV2 binds into the cell, the virus/plasma membrane receptor can be monitored as an internalization process mediated by virus particles (Fig. 7a). For this purpose, we produced lentivirus expressing the SARS-CoV2 spike protein[52]. We then added the viral suspension containing the SARS-CoV2 spike protein to cells over expressing the ACE2 receptor covalently linked to the small domain of NLuc. This design strategy enabled the simulation of the SARS-CoV2 infection in real-time and in living cells. Our results demonstrate that we can monitor membrane protein trafficking of SARS-CoV2 spike protein with a good signal-noise ratio and where the entire infection process takes approximately three hours and this virus–receptor complex continues to be internalized for some time thereafter. As such, we believe this assay will allow us to study neutralizing antibodies or antivirals. Furthermore, this technology can also be extended to other infection systems using other viruses like those mediated by a GPCR, as in the case of the CCR5 and CXCR4, members of the chemokine receptor family, during an HIV-1 infection[9,53].

In summary, the main advantage of our method is that the internalization of the receptor can be monitored in real-time and the versatility to study a wide range of internalization mechanisms, ranging from GPCRs, RTKs, antibody to viral entry into the cell. Our assay shows a dynamic range between 1.5 and 3, which was similar to other approaches using the structural complementation of NLuc[29,54] and slightly lower as compared to other approaches measuring receptor-arrestin interactions in real-time[55]. Some aspects remain to be studied in our methodology, such as to evaluate whether if tagging the receptors with the SmBiT alters its native pharmacology and distribution at the plasma membrane, as well as expressing the tagged receptors at endogenous expression levels.

In conclusion, we both theoretically and experimentally illustrated the versatility of our structural complementation approach to study a wide range of membrane protein internalization mechanisms. Furthermore, we defined a comprehensive set of cell-based assays and demonstrated their ability of monitoring membrane protein trafficking. These assays have direct application to drug discovery and drug development. The internalization rates vary dramatically ranging from a few minutes (in the case of GPCRs) to a few hours (in the case of viral entry into the cells). Altogether, a universal methodology such as that illustrated in this work will accelerate drug discovery and drug development for numerous types of diseases where membrane protein trafficking plays a key role.

## Methods

The ability to monitor internalization in response to ligand stimulation was assessed in HEK293 cells (American Type Culture Collection, Cat. No. CRL-1573) expressing the corresponding human receptor. In the case of GPCRs, the assay was performed using the corresponding endogenous ligands (i.e., epinephrine).

**Materials**. Cell culture medium and cell culture additives were from Promega and Life Technologies.

**Chemicals and peptides**. All chemicals were obtained from Sigma-Aldrich (St. Louis, MO, USA) unless otherwise stated. The restriction enzymes were obtained from New England Bio Labs (Ipswich, MA, USA). All ligand peptides were synthesized by AnyGen (Gwangju, Korea). The synthesized peptide purity was greater than 98% as determined by high-performance liquid chromatography analysis. All peptides were dissolved in dimethyl sulfoxide and then diluted in media to the desired working concentrations.

**NanoBit Technology**. The NanoBit starter kit containing the plasmids and the necessary reagents for the development of the structural complementation assays used in this study was obtained from Promega Company (Madison, Wisconsin, USA).

**Primer design**. We designed primers to introduce genes of interest into pBiT1.1-C [TK/LgBiT], pBiT2.1-C [TK/SmBiT], pBiT1.1-N [TK/LgBiT] and pBiT2.1-N [TK/SmBiT] vectors (Supplementary Table 1). We selected at least one of these three sites as one of the two unique restriction enzymes needed for directional cloning due to the presence of an in-frame stop codon that divides the multicloning site (Supplementary Figs. 3, 4). We incorporated nucleotide sequences into the primers as shown in Supplementary Table 1 to encode the linker residues shown in Supplementary Figs. 1, 2. For pBiT1.1-C [TK/LgBiT] and pBiT2.1-C [TK/SmBiT] vectors, we make sure that the 5′ primer contained an ATG codon and a potent Kozak consensus sequence (GCCGCCACC). For pBiT1.1-N [TK/LgBiT] and pBiT2.1-N [TK/SmBiT] vectors, we ensured that the 3′ primer contained a stop codon.

**Cloning**. We prepared a 1% agarose gel to run the digested DNA plasmid and insert and proceed to cut the corresponding bands. Once the corresponding vector and insert bands were purified, we determined the DNA concentration using a spectrophotometer. We performed DNA ligation to fuse the insert to the recipient plasmid. We prepared ligation reactions of around 100 ng of total DNA including 50 ng of plasmid vector. We set up a recipient plasmid-insert ratio of approximately 1:3. We also set up negative controls in parallel. For instance, ligation of the recipient plasmid DNA without any insert provided information about how much background of undigested or self-ligating recipient plasmid was present.

**Isolation of the positive clones**. We picked 3–10 individual bacterial colonies and transferred them into 1 mL of LB medium containing ampicillin (100 μg/mL) and incubated them for 6 h. Then, we used 200 μL of bacterial suspension and transferred it to 5 mL of LB medium containing the same concentration of ampicillin and incubated overnight at 37.5 °C with shaking at 200 rpm. We performed miniprep DNA purifications using 5 mL of LB grown overnight following the manufacturer's instructions (Life Technologies). To identify successful ligations, we set up PCR reactions using the DNA obtained from mini-preps as a template with the same primers as during the first PCR used for cloning. Positive clones produced the PCR products with the corresponding insert size. We verified the construct sequence by sequencing using the primers shown in Supplementary Table 1.

**Proximity ligation assay (PLA)**. HEK293 cells were seeded in an 8 well Lab-Tek II Chamber Slide (Life Technologies) with a density of $2 \times 10^5$ cells per well. The next morning, cells were transfected with 200 ng of β2AR-SmBiT and 200 ng of LgBiT-FYVE constructs using Viafect (Promega Corporation). The next day, samples were treated with 10 μM epinephrine final concentration for 5 min, and immediately after, cells were incubated with 4% paraformaldehyde for 15 min at room

temperature. After the cells were rinsed with PBS and permeabilized with PBS containing 0.1% Tween 20 (PBST). Then, cells were incubated with blocking buffer (Duolink blocking buffer for PLA) at 37.5 °C for 1 h and followed by incubation with the primary antibodies, anti-β2AR (ThermoFisher catalog number MA5-38441), anti-NLuc (Promega catalog number N7000), or anti-Early Endosome Antigen 1 (Abcam, catalog number ab109110) at 1:1000 by diluting in the Duolink antibody dilution buffer at 4 °C overnight. After, three washes (5 min each) with PBST, the cells were incubated with PLA probes (PLUS and MINUS PLA probes) in a pre-heated humidity chamber for 1 h at 37.5 °C. Three washes (5 min each) were then performed. Ligation of the two PLA probes was performed by incubation of the slides in a pre-heated humidity chamber for 30 min at 37 °C in the presence of ligase and 1× ligation buffer. Confocal images were taken using a Nikon A1R-s Confocal Microscope using a 40× Plan Fluor/0.75 NA objective.

**Widefield bioluminescence microscopy.** Bioluminescence imaging was performed using an Olympus LV200 wide field inverted microscope, equipped with a 60×/1.42NA oil immersion objective lens and 0.5× tube lens. HEK293 cells were seeded into a 35 mm dish (Manufacturer) with a density of $4 \times 10^5$ cells per well. The next day the cells were transfected using 1 μg of β2AR-SmBiT plus 1 μg of LgBiT-FYVE and ViaFect™ Transfection Reagent (Promega, catalog number E4981). Twenty-four hours later, on the day of imaging, medium was removed, and cells were incubated with 1 mL of Opti-MEM containing furimazine for 10 min at 37 °C before epinephrine (10 μM final concentration) was added and allowed to equilibrate for 5 min at 37 °C. Luminescence images were taken by capturing total luminescence for (120 s exposure time).

**Internalization assay using NanoBit Technology.** HEK293 cells were maintained in Dulbecco's modified Eagle's medium (DMEM) supplemented with 10% fetal bovine serum (FBS), 100 U/ml penicillin G, and 100 μg/ml streptomycin (Invitrogen; Carlsbad, CA, USA). At 1 day before transfection, the cells were seeded in 96-well plates at a density of $2.5 \times 10^4$ cells per well. A mixture containing 100 ng receptor construct containing the LgBit or SmBit and 50 ng of the Endofin domain containing one of the two domains of Nano luciferase and 0.3 μl Viafect (Promega) was prepared and added to each well. We tested four Endofin-receptor spatial orientations. The one with the highest signal was chosen for further experiments to obtain maximum sensitivity. At 24 h post-transfection, the medium was aspirated and replaced with 100 μl OPTIMEM (Life Technologies, Grand Island, NY, USA). After a 10 min incubation, 25 μl substrate (furimazine) was added, and once every minute subsequent luminescence measurements were taken for 5–10 min for signal stabilization. A total of 10 μl of ligand, antibody, or viral suspension was then added to each well and luminescence measurements were recorded immediately and once every minute for 1–3 h (Synergy 2 Multi-Mode Microplate Reader Bio-Tek, Winooski, VT, USA). Fold induction was calculated by normalizing the luminescent signal against vehicle.

**Lentivirus-mediated expression of the spike protein of SARS-CoV2.** All manipulations were taking place in a biosafety cabinet at all times. HEK293 cells were transfected with the plasmids containing SARS-CoV-2, Wuhan-Hu-1 (GenBank: NC_045512), spike-pseudotyped lentiviral kit (NR-52948, from Bei Resources) designed to generate pseudotyped lentiviral particles expressing the spike (S) glycoprotein gene, as well as luciferase (Luc2) and green fluorescent protein (GFP). Seventy-two hours after transfection, the medium was collected in a 50 ml tube and store at −80 °C for further applications. This protocol only requires Biosafety Level 1 (BSL1) conditions and the viruses used in this protocol were replication-defective.

**Statistics and reproducibility.** The results were analyzed using the Prism 7 application (Graph Pad Software Inc., San Diego, CA). Dose–response curves were fitted using the following three-parameter equation:

$$\text{Response} = \text{Bottom} + \frac{\text{Top} - \text{Bottom}}{1 + 10^{(\log \text{EC}_{50} - \log[A])}} \qquad (1)$$

where Bottom and Top are the lower and upper plateaus, respectively, of the concentration–response curve, [A] is the molar concentration of the agonist, and $\text{EC}_{50}$ is the molar concentration of agonist required to generate a response halfway between the top and the bottom.

Z-factors expressing the high-throughput suitability were calculated with the following equation:

$$Z = 1 - \left(3\sigma_S + 3\sigma_C\right)\left(\mu_S - \mu_C\right) \qquad (2)$$

where $\sigma_S$ and $\sigma_C$ are the standard deviations of fold induction. $\mu_S$ and $\mu_C$ express the mean of fold induction values of positive and negative control, respectively. As a positive control, we defined the samples treated with the corresponding ligand. Medium OptiMEM was used as a negative control in all Z factor experiments.

**Reporting summary.** Further information on research design is available in the Nature Research Reporting Summary linked to this article.

## Data availability
Data set corresponding to dose–response curves and Z′ factors for each assay is available for download at Dryad (https://doi.org/10.5061/dryad.7d7wm37x0). Some plasmids have been deposited in Addgene (Deposit number 80697). Additional data that support the findings of this study are available in supplementary data and from the corresponding author upon request.

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

## Acknowledgements

Research supported in this manuscript was supported in part by the National Heart, Lung, and Blood Institute (NHLBI) of the National Institutes of Health (NIH) under award number R15 HL141963 (B.K.M.), the American Heart Association (AHA) under award number 18AIREA 33960175 (B.K.M.), and a grant from the Robert J. Kleberg, Jr. and Helen C. Kleberg Foundation (B.K.M.). We are also grateful for grant support from the University of Houston; Grants to Enhance Research on COVID-19 and the Pandemic (B.K.M.). The funders had no role in the preparation of the manuscript or decision to publish this manuscript.

## Author contributions

A.R.A. and B.K.M. conceived and designed the research; A.R.A., E. Y. L., G.-R., and E.A.M. conducted experiments; A.R.A., J.Y.S., and B.K.M. contributed new reagents; A.R.A. and B.K.M. developed analytic tools and technology; A.R.A., E.A.M., R.A.B., and B.K.M. performed data analysis and interpreted the results; A.R.A., R.A.B., and B.K.M. wrote or contributed to the writing of the manuscript.

## Competing interests

The authors declare no competing interests.
