## [Peer Review File · Communications Biology]

Reviewers' comments:

Reviewer #1 (Remarks to the Author):

Reyez-Alcaraz et al. present in this article a novel technology for monitoring membrane protein trafficking by bioluminescence. Their approach is based on the split nanoluciferase. The small fragment of the split nano luciferase is fused to the C-terminal of the membrane receptor, while the large fragment is fused to the FYVE domain of the human Endofin, which selectively binds phosphatidylinositol 3-phosphate from early endosome. Receptor internalization leads to complementation of the nano luciferase on the early endosome and thus to a bioluminescent response. The authors demonstrate the use of their methods by monitoring different internalization mechanism such as receptor internalization/recycling, antibody-mediated internalization and SARS-CoV2 viral entry.

Remarks and comments.

1. The authors claim in their introduction that their approach allows one to "quantify in real time membrane protein internalisation and recycling at endogenous expression levels". The experiments shown in the paper have been done in HEK cells by transient transfection, which leads to expression levels far from endogenous levels. To demonstrate the usability of their approach with endogenous expression levels, the authors should use CRISPR edited receptors, or alternative methods allowing expression of tagged proteins at endogenous levels. Along this line, a full characterisation of the expression levels of the different fusion proteins would be useful to evaluate the minimal expression level necessary to see a significant signal.

2. The authors assert that the complementation of Nanoluc takes place in the early endosome. No experiment demonstrates this assertion explicitly. It would be useful to demonstrate that the complementation occurs indeed in early endosomes.

3. The bioluminescence increases upon internalisation shown in the paper are between 1.5 to 3 fold. The authors should comment whether it is a high or low dynamic range compared to other techniques, and whether it is efficient enough to work at lower expression levels (see also comment 1).

4. The signal of luciferase is known to decay upon consumption of furimazine. Is the signal of Nanoluc constant during the time window of the presented experiments? If no, how this decay affect the apparent kinetics of internalisation ?

5. Nanoluc can be imaged with a bioluminescence imager. Is the technology compatible with bioluminescence imaging ? This would be useful to obtain spatial information on receptor internalisation.

Overall, the paper is pleasant to read. The idea of generating selectively a bioluminescent signal when the receptor is in the early endosome to monitor receptor internalisation is very elegant. However, the authors chose the brief communication format, which makes the article a bit to concise according to me. As a reader and potential user, I would have appreciated more explanations about the rational of the tool design and more experimental details. In its current state, I find the study a bit superficial. A deeper evaluation of the robustness and limits of the approach would strengthen the study, and better demonstrate the potential of the approach.

I think a major revision is necessary for this study being suited for Communication Biology. Otherwise, in the current sate, I would recommend publication in Scientific Reports.

Reviewer #2 (Remarks to the Author):

In this manuscript, Reyes-Alcaraz et al develop an internalization assay based on the complementation of receptors or membrane proteins (GPCR, HER2, FAM19A5, SARS-CoV2 spike) with the FYVE domain of an endosome protein endofin. Their assay employs the well-known

NanoBit complementation technology (Promega) and a BRET-based proximity readout. Although interesting, particularly in light of its utility across membrane-protein classes, the authors' approach cannot really be considered novel, but rather as an addition to the available real-time internalization assay battery, e.g. FRET, TR-FRET, BRET approaches. This is not necessarily a problem, but needs to be openly acknowledged from the outset. Indeed, recent publications (all since 2016) using NanoBit include PMIDs: 26569370, 29309765, 30987329, 31160049, 31467080, 31583945, 32381507, 32807782, 32053779. Unfortunately, these studies, together with the authors' own studies using this technology (PMID: 30272007 and 33292475) were not even discussed or mentioned in the introduction or discussion to place the work in context. Instead, there are several vague generalisations e.g. 'Receptor internalization is one part of cell signalling...' and others that require updating e.g. 'Current technologies that monitor internalization are costly and limited to a specific internalization mechanism.' This may have been true in 2004, 2001 and 2012 when the cited references were written.

The data presented are very brief and could be made more convincing with additional characterisation of their assays, for example concentration-response curves rather than single ligand concentration additions and Z prime analyses to assess assay robustness and reproducibility. In this regard, the authors should perhaps comment of the small dynamic range and signal-to-noise ratios which may affect the utility of their assays.

I realise that this is written as a short communication, but there needs to be considerably more detail in the methods. There were no statistical analyses performed. Given the level of methodological detail currently provided (particularly about the design and construction (cloning?) of their constructs, it would be very difficult for other researchers to reproduce this work.

Finally, as has been touched upon earlier, the discussion section should include weighing up the strengths and weaknesses of this assay in the context of the literature. Equally, it must be acknowledged that this assay fundamentally relies on transfected receptors/complementation partners and does not measure endogenously expressed receptors under physiological conditions as is claimed.

Reviewer #3 (Remarks to the Author):

This brief paper reports an assay to follow ligand-stimulated endocytosis of cell surface proteins. The authors make use of the NanoBit split luciferase system, and express one part of the luciferase as a fusion to the cytoplasmic tail of cell surface proteins, and the other part as a fusion to a domain that is targeted to endosomes. When the cell surface protein is endocytosed it reaches endosomes, the luciferase assembles, and its activity can be detected by bioluminescence. The authors demonstrate the utility of this method by applying it to several G-protein coupled receptors, as well as the EGF receptor and the SARS-CoV-2 receptor ACE2.

The advantage of the assay is that allows real-time, rather than end point, assay of endocytosis. There are of course many other assays of endocytosis, but since it is an important and widely studied process that is linked to many receptor activation events, then another assay may well be of interest.

However, to make this work suitable for publication the authors need to provide more detail of the experiments and constructs. In addition, then need to characterize the system more thoroughly. These changes are essential to allow others to reproduce and apply the method, and to allow users to be confident of what they are measuring.

1) The authors need to provide sequences to show exactly how the parts of luciferase were attached to the receptors. The authors also need to provide the sequence and description of how the endofin FYVE domain is attached to LgBiT. Indeed, the description of this construct in the text is confusing – presumably they did not synthesis the FYVE domain, but rather ordered DNA encoding the FYVE domain and cloned it into a plasmid next to LgBiT.

2) The authors should show immunofluorescence images to confirm that the endofin-LgBiT fusion is localized to early endosomes.

3) Early endosomes are NOT "composed mostly of PtdIns 3 4 5-trisphosphate". A few percent of the lipid content of early endosomes is the lipid PtdIns 3-phosphate.

4) The experiment with the pseudovirus and ACE2 needs a negative control (either nothing added, or a pseudovirus without SARS-CoV-2 spike), to measure the constitutive rate of endocytosis of ACE2 in the absence of virus.

5) The authors need to add a clear caveat that attaching SmBiT to the cytoplasmic tail of a cell surface protein could alter its delivery to the surface or its endocytosis.

6) The authors should discuss the advantages and disadvantages of their method compared to previous assays for endocytosis. They should also mention and cite two other papers that use split luciferase in a different way to follow endocytosis (PubMed IDs 32053779 and 31583945).

7) Do the authors intend to make the plasmids used in the paper freely available at Addgene? This would be appreciated by others, and if this is their intention, it should be stated.

We thank the Reviewers for their insightful comments which have led to a significantly improved the manuscript.

Reviewer #1 (Remarks to the Author):

Specific remarks and comments.

#1. The authors claim in their introduction that their approach allows one to "quantify in real time membrane protein internalisation and recycling at endogenous expression levels". The experiments shown in the paper have been done in HEK cells by transient transfection, which leads to expression levels far from endogenous levels. To demonstrate the usability of their approach with endogenous expression levels, the authors should use CRISPR edited receptors, or alternative methods allowing expression of tagged proteins at endogenous levels. Along this line, a full characterisation of the expression levels of the different fusion proteins would be useful to evaluate the minimal expression level necessary to see a significant signal.

We completely agree with the Reviewer's comment. Generating the CRISPR edited receptors is taking us much longer than anticipated. Since the information of our manuscript is time-sensitive, we have decided to modify the description of our work and eliminate the claim that our methodology can be used with endogenous expression levels, and we highlight that our methodology is based on transient transfection in the discussion section (lines 258-267).

2. The authors assert that the complementation of Nanoluc takes place in the early endosome. No experiment demonstrates this assertion explicitly. It would be useful to demonstrate that the complementation occurs indeed in early endosomes.

Thank you for your suggestion. To demonstrate that the luminescent signal is generated from the early endosome, we now include data using the technique of "Proximity Ligation Assay" (Figure 3, lines 145-162). With this approach, we demonstrate that the receptor and the FYVE domain-containing NLuc are localized at the early endosome within a few minutes after ligand treatment.

3. The bioluminescence increases upon internalisation shown in the paper are between 1.5 to 3-fold. The authors should comment whether it is a high or low dynamic range compared to other techniques, and whether it is efficient enough to work at lower expression levels.

As suggested by the Reviewer, we have added a discussion of this observation that including a statement that the dynamic range is similar or only slightly lower compared to similar approaches previously reported (lines 324-330). As previously mentioned in comment 1 above, the gene-editing experiments for some GPCRs are taking longer than we

expected, and we clarify that the current state of our methodology relies only on transient transfections.

4. The signal of luciferase is known to decay upon consumption of furimazine. Is the signal of Nanoluc constant during the time window of the presented experiments? If no, how this decay affect the apparent kinetics of internalisation?

Yes, we were aware of this decay in the signal caused by substrate depletion. To control this variable we normalized the kinetic curves with the corresponding vehicle-treated samples.

5. Nanoluc can be imaged with a bioluminescence imager. Is the technology compatible with bioluminescence imaging? This would be useful to obtain spatial information on receptor internalisation.

Yes, following the Reviewer's suggestion we discovered that our methodology is also compatible with live cell imaging. We now describe this result (Figure 4, lines 165-177).

Reviewer #2 (Remarks to the Author):

1. In this manuscript, Reyes-Alcaraz et al develop an internalization assay based on the complementation of receptors or membrane proteins (GPCR, HER2, FAM19A5, SARS-CoV2 spike) with the FYVE domain of an endosome protein endofin. Their assay employs the well-known NanoBit complementation technology (Promega) and a BRET-based proximity readout. Although interesting, particularly in light of its utility across membrane-protein classes, the authors' approach cannot really be considered novel, but rather as an addition to the available real-time internalization assay battery, e.g. FRET, TR-FRET, BRET approaches. This is not necessarily a problem but needs to be openly acknowledged from the outset. Indeed, recent publications (all since 2016) using NanoBit include PMIDs: 26569370, 29309765, 30987329, 31160049, 31467080, 31583945, 32381507, 32807782, 32053779. Unfortunately, these studies, together with the authors' own studies using this technology (PMID: 30272007 and 33292475) were not even discussed or mentioned in the introduction or discussion to place the work in context. Instead, there are several vague generalisations e.g. 'Receptor internalization is one part of cell signalling...' and others that require updating e.g. 'Current technologies that monitor internalization are costly and limited to a specific internalization mechanism.' This may have been true in 2004, 2001 and 2012 when the cited references were written.

Thank you for your comment. We acknowledged that our methodology is based on a combination of NanoBit technology and a BRET-based proximity readout (lines 59-73) and

a long the entire manuscript we described our assay as a viable alternative to current methodologies. Also, We have now referenced most of the suggested publications. We also added discussion (lines 216-233) to better place our work in its proper context. Also, we have rephrased the sentences and updated the corresponding references (lines 46-57).

2. The data presented are very brief and could be made more convincing with additional characterisation of their assays, for example concentration-response curves rather than single ligand concentration additions and Z prime analyses to assess assay robustness and reproducibility. In this regard, the authors should perhaps comment of the small dynamic range and signal-to-noise ratios which may affect the utility of their assays.

These are excellent suggestions. Following the Reviewer's suggestion, we now include concentration-response curves and have calculated the internalization potencies (Figures 1 and 6). We have now also assessed the robustness of each assay by calculating Z' factors (Figure legends of each Figure). We also discuss the dynamic range of our technology compared to similar approaches (Discussion section, lines 221-233).

3. I realise that this is written as a short communication, but there needs to be considerably more detail in the methods. There were no statistical analyses performed. Given the level of methodological detail currently provided (particularly about the design and construction (cloning?) of their constructs, it would be very difficult for other researchers to reproduce this work.

We agree with the Reviewer and have now added more details in the Material & Methods, particularly about cloning (lines 373-411 and Supplementary Table I).

Finally, as has been touched upon earlier, the discussion section should include weighing up the strengths and weaknesses of this assay in the context of the literature. Equally, it must be acknowledged that this assay fundamentally relies on transfected receptors/complementation partners and does not measure endogenously expressed receptors under physiological conditions as is claimed.

Please see response #1 to Reviewer #1 above. We acknowledge and explain that our assay relies on transient transfections and not endogenously expressed receptors (lines 258-267).

Reviewer #3 (Remarks to the Author):

This brief paper reports an assay to follow ligand-stimulated endocytosis of cell surface proteins. The authors make use of the NanoBit split luciferase system, and express one part of the luciferase as a fusion to the cytoplasmic tail of cell surface proteins, and the other

part as a fusion to a domain that is targeted to endosomes. When the cell surface protein is endocytosed, it reaches endosomes, the luciferase assembles, and its activity can be detected by bioluminescence. The authors demonstrate the utility of this method by applying it to several G-protein coupled receptors, as well as the EGF receptor and the SARS-CoV-2 receptor ACE2.

The advantage of the assay is that allows real-time, rather than end point, assay of endocytosis. There are of course many other assays of endocytosis, but since it is an important and widely studied process that is linked to many receptor activation events, then another assay may well be of interest.

However, to make this work suitable for publication the authors need to provide more detail of the experiments and constructs. In addition, then need to characterize the system more thoroughly. These changes are essential to allow others to reproduce and apply the method, and to allow users to be confident of what they are measuring.

1. The authors need to provide sequences to show exactly how the parts of luciferase were attached to the receptors. The authors also need to provide the sequence and description of how the endofin FYVE domain is attached to LgBiT. Indeed, the description of this construct in the text is confusing – presumably they did not synthesis the FYVE domain, but rather ordered DNA encoding the FYVE domain and cloned it into a plasmid next to LgBiT.

We thank the Reviewer for their positive comments and specific points, We have added details to the Methods and have now provided a detailed description of the sequences, where is illustrated about how the FYVE domain is attached to LgBiT (Supplementary Figures 1 and 2).

2. The authors should show immunofluorescence images to confirm that the endofin-LgBiT fusion is localized to early endosomes.

Please see Response #2 to Reviewer 1 above. To demonstrate that the luminescent signal is generated from the early endosome we used the technique “Proximity Ligation Assay” (Figure 3, lines 145-162). With this approach, we verified that the receptor and the FYVE domain-containing NLuc are localized at the early endosome a few minutes after ligand treatment.

3. Early endosomes are NOT “composed mostly of PtdIns 3 4 5-trisphosphate”. A few percent of the lipid content of early endosomes is the lipid PtdIns 3-phosphate.

Thank you for sharing this information. We have removed the sentence regarding the composition of the early endosomes.

4. The experiment with the pseudovirus and ACE2 needs a negative control (either nothing added, or a pseudovirus without SARS-CoV-2 spike), to measure the constitutive rate of endocytosis of ACE2 in the absence of virus.

We followed the Reviewer's suggestion and we now include data with the corresponding negative control (only vehicle added, Figure 7).

5. The authors need to add a clear caveat that attaching SmBiT to the cytoplasmic tail of a cell surface protein could alter its delivery to the surface or its endocytosis.

As suggested by the Reviewer, we discuss the possibility that attaching the SmBiT to the tail of the receptor could alter its delivery to the cell surface and/or its native internalization (327-330).

6. The authors should discuss the advantages and disadvantages of their method compared to previous assays for endocytosis. They should also mention and cite two other papers that use split luciferase in a different way to follow endocytosis (PubMed IDs 32053779 and 31583945).

The advantages and disadvantages of our methodology are included in the Discussion section (lines 321-330).

7. Do the authors intend to make the plasmids used in the paper freely available at Addgene? This would be appreciated by others, and if this is their intention, it should be stated.

Yes, we have now added that we will make all our plasmids freely available at Addgene within the next few weeks.

Reviewers' comments:

Reviewer #1 (Remarks to the Author):

Reyez-Alcaraz et al. present in this article a novel technology for monitoring membrane protein trafficking by bioluminescence. Their approach is based on the split nanoluciferase. The small fragment of the split nano luciferase is fused to the C-terminal of the membrane receptor, while the large fragment is fused to the FYVE domain of the human Endofin, which selectively binds phosphatidylinositol 3-phosphate from early endosome. Receptor internalization leads to complementation of the nano luciferase on the early endosome and thus to a bioluminescent response. The authors demonstrate the use of their methods by monitoring different internalization mechanism such as receptor internalization/recycling, antibody-mediated internalization and SARS-CoV2 viral entry. The idea of generating selectively a bioluminescent signal when the receptor is in the early endosome to monitor receptor internalisation is very elegant.

The manuscript have been modified according to most reviewers' comments. The authors have included a longer discussion, which enables to better appreciate the originality of the work. The insertion of imaging data on Fig 4 showing that bioluminescence imaging can be used to visualize internalization is a nice addition, as it provides additional spatial information. The authors present also a proximity ligation assay to show that the complementation of nanoluc occurs in the early endosomes. The fluorescent signal is however quite weak and present in all cells. I would have expected punctuate signals only in a subpopulation of cells as cells are transiently transfected. The use of higher magnification to improve spatial resolution would have been also relevant here. The authors should better explain how they interpreted this experiment and drew their conclusions. Overall, the manuscript has been improved, and can be suitable for publication if the above remark is addressed.

Reviewer #2 (Remarks to the Author):

In this substantially revised and expanded manuscript, Reyes-Alcaraz et al develop an internalization assay based on the complementation of receptors or membrane proteins (GPCR, HER2, FAM19A5, SARS-CoV2 spike) with the FYVE domain of an endosome protein endofin. Although the authors should be acknowledged for making efforts to place their work in its appropriate context in terms of the published literature, there are still numerous issues with the manuscript:

1. There are multiple instances where the writing style/phrasing is vague and even obscures the meaning. E.g. (line 44-47) "More specifically, receptor trafficking studies follow the functional process of receptor binding to agonists. Receptor internalization is vital in a wide variety of physiological functions and disease states such as neurological disorders¹, viral infection², cardiomyopathies³, cancer⁴ among others."
2. Despite its potential utility as a signaling assay, it is important not to overstate the significance of this work given that this is an artificial assay with overexpressed receptors – claims should be toned down especially with regard to therapeutic development e.g. (line 69-73) "...this method can universally be applied to monitor the internalization of a wide range of membrane proteins and be used in the elucidation of novel molecular mechanisms as well as the development of therapeutic agents for cancer, cardiopulmonary and infectious diseases including COVID-19."
3. The methods are still insufficiently detailed e.g. there is no mention of how the Z' analysis was conducted, nor is there any description of the luminescent imaging or information about 'antibody A' and 'B'.
4. Some of the explanations of the data are confusing and do not match the data presented. E.g. Figure 4 – the decrease in luminescence intensity should be quantified as it is not clear as presented. Relating to Figure 6C (line 203-5) "We achieved to quantify and compare the internalization potencies for the two antibodies, highlighting that antibody A was slightly more potent than antibody B (Figure 6C)." The concentration-response curves have near identical potency, and the data points have no error bars. A Z' of 0.94 is unbelievably high. Which data was used for this calculation?
5. Again, the discussion/conclusion contains generalizations that are not explicitly supported by

the data or the literature and should be toned down/rephrased and appropriately referenced. E.g. "After studying different physiological processes where membrane receptors are involved, we conclude that almost any membrane protein expressed at the cell surface undergoes internalization..."

6. There are formatting errors throughout the references.

Reviewer #3 (Remarks to the Author):

The authors have done a good job in revising this paper by adding extra data, extra information and extra discussion to ensure that others will be able to evaluate the method and successfully apply if they wish. Apart from one very minor suggestion (see below), I am happy to recommend acceptance.

Suggestion: Figures 3B, 3E and Figure 4, have single colour channels that are quite faint and so hard to see. It would be better to show these as grey scale, with only merged images in colour.

We thank the Reviewers for their insightful comments which have led to a significantly improved the manuscript.

Reviewer #1 (Remarks to the Author):

Reyez-Alcaraz et al. present in this article a novel technology for monitoring membrane protein trafficking by bioluminescence. Their approach is based on the split nanoluciferase. The small fragment of the split nano luciferase is fused to the C-terminal of the membrane receptor, while the large fragment is fused to the FYVE domain of the human Endofin, which selectively binds phosphatidylinositol 3-phosphate from early endosome. Receptor internalization leads to complementation of the nano luciferase on the early endosome and thus to a bioluminescent response. The authors demonstrate the use of their methods by monitoring different internalization mechanism such as receptor internalization/recycling, antibody-mediated internalization and SARS-CoV2 viral entry. The idea of generating selectively a bioluminescent signal when the receptor is in the early endosome to monitor receptor internalisation is very elegant.

The manuscript have been modified according to most reviewers' comments. The authors have included a longer discussion, which enables to better appreciate the originality of the work. The insertion of imaging data on Fig 4 showing that bioluminescence imaging can be used to visualize internalization is a nice addition, as it provides additional spatial information. The authors present also a proximity ligation assay to show that the complementation of nanoluc occurs in the early endosomes. The fluorescent signal is however quite weak and present in all cells. I would have expected punctuate signals only in a subpopulation of cells as cells are transiently transfected. The use of higher magnification to improve spatial resolution would have been also relevant here.

Thanks so much for your observation. We tried to improve the fluorescent signal of our images by using another confocal microscope (Nikon A1R-s Confocal Microscope) to have better sensitivity and more quality in our figures. By analyzing other cells populations, we were able to observe punctuate signals typically seen in GPCR internalization.

Reviewer #2 (Remarks to the Author):

1. There are multiple instances where the writing style/phrasing is vague and even obscures the meaning. E.g. (line 44-47) "More specifically, receptor trafficking studies follow the functional process of receptor binding to agonists. Receptor internalization is vital in a wide variety of physiological functions and disease states such as neurological disorders¹, viral infection², cardiomyopathies³, cancer⁴ among others."

We completely agree with the Reviewer's comments. We have rearranged those sentences (please see lines 46-61). We included more specific information and explained with more clarity the background that supports our work.

2. Despite its potential utility as a signaling assay, it is important not to overstate the significance of this work given that this is an artificial assay with overexpressed receptors – claims should be toned down especially with regard to therapeutic development e.g. (line 69-73) “...this method can universally be applied to monitor the internalization of a wide range of membrane proteins and be used in the elucidation of novel molecular mechanisms as well as the development of therapeutic agents for cancer, cardiopulmonary and infectious diseases including COVID-19.”

Thank you for your suggestion. We followed your suggestions and we toned-down and modified the text of our manuscript (please see lines 104-106) considering the potential utility of our signaling assay.

3. The methods are still insufficiently detailed e.g. there is no mention of how the Z' analysis was conducted, nor is there any description of the luminescent imaging or information about 'antibody A' and 'B'.

As suggested by the reviewer, we included and explained how the dose response curves were generated and how the Z' analysis was conducted (please see lines 529-545) and the conditions about how the luminescent imaging was performed (please see lines 487-498).

4. Some of the explanations of the data are confusing and do not match the data presented. E.g. Figure 4 – the decrease in luminescence intensity should be quantified as it is not clear as presented. Relating to Figure 6C (line 203-5) “We achieved to quantify and compare the internalization potencies for the two antibodies, highlighting that antibody A was slightly more potent than antibody B (Figure 6C).” The concentration-response curves have near identical potency, and the data points have no error bars. A Z' of 0.94 is unbelievably high. Which data was used for this calculation?

We agree with the reviewer and have now added data regarding the decrease in bioluminescence, presumably due to substrate depletion was 25% in a time frame of 14 minutes after ligand treatment (please see histogram graphs in Supplementary Figure 6 and lines 215-217).

Regarding the potencies for the FAM19A5 antibodies A and B, we changed the sentence and instead we highlighted there is no statistical difference between the two antibodies (Please see lines 244-245).

We also provide data about how we extended the Z' calculations to 96 data points (Please see Supplementary Figure 5).

5. Again, the discussion/conclusion contains generalizations that are not explicitly supported by the data or the literature and should be toned down/rephrased and appropriately referenced. E.g. “After studying different physiological processes where membrane receptors are involved, we conclude that almost any membrane protein

expressed at the cell surface undergoes internalization...”

We rephrased and modified the sentence. We toned down some expressions in all the text and appropriately added the corresponding references (please see lines 274-286).

6. There are formatting errors throughout the references.

We carefully revised the references one by one following the instructions for Authors of this Journal.

Reviewer #3 (Remarks to the Author):

The authors have done a good job in revising this paper by adding extra data, extra information and extra discussion to ensure that others will be able to evaluate the method and successfully apply if they wish. Apart from one very minor suggestion (see below), I am happy to recommend acceptance.

Suggestion: Figures 3B, 3E and Figure 4, have single colour channels that are quite faint and so hard to see. It would be better to show these as grey scale, with only merged images in colour.

Thanks for this observation. This time we used a different confocal microscope (Nikon A1R-s Confocal Microscope) and we were able to improve the fluorescent signal of our pictures for a better visualization.

REVIEWERS' COMMENTS:

Reviewer #1 (Remarks to the Author):

The revised manuscript is significantly improved, and suitable for publication.